# Zhamanshin astrobleme provides evidence for carbonaceous chondrite and post-impact exchange between ejecta and Earth's atmosphere

Tomáš Magna[1], Karel Žák[2], Andreas Pack[3], Frédéric Moynier[4,5], Bérengère Mougel[4], Stefan Peters[3], Roman Skála[2], Šárka Jonášová[2], Jiří Mizera[6] & Zdeněk Řanda[6]

Chemical fingerprints of impacts are usually compromised by extreme conditions in the impact plume, and the contribution of projectile matter to impactites does not often exceed a fraction of per cent. Here we use chromium and oxygen isotopes to identify the impactor and impact-plume processes for Zhamanshin astrobleme, Kazakhstan. $\varepsilon^{54}Cr$ values up to 1.54 in irghizites, part of the fallback ejecta, represent the $^{54}Cr$-rich extremity of the Solar System range and suggest a CI-like chondrite impactor. $\Delta^{17}O$ values as low as $-0.22$‰ in irghizites, however, are incompatible with a CI-like impactor. We suggest that the observed $^{17}O$ depletion in irghizites relative to the terrestrial range is caused by partial isotope exchange with atmospheric oxygen ($\Delta^{17}O = -0.47$‰) following material ejection. In contrast, combined $\Delta^{17}O$-$\varepsilon^{54}Cr$ data for central European tektites (distal ejecta) fall into the terrestrial range and neither impactor fingerprint nor oxygen isotope exchange with the atmosphere are indicated.

[1] Czech Geological Survey, Klárov 3, Prague 1 CZ-118 21, Czech Republic. [2] Institute of Geology of the Czech Academy of Sciences, v.v.i., Rozvojová 269, Prague 6 CZ-165 00, Czech Republic. [3] Geowissenschaftliches Zentrum, Abteilung Isotopengeologie, Universität Göttingen, Goldschmidtstraße 1, Göttingen D-37077, Germany. [4] Institut de Physique du Globe de Paris, Université Paris Diderot, 1 rue Jussieu, Paris F-75005, France. [5] Insitut Universitaire de France, Paris F-75005, France. [6] Nuclear Physics Institute of the Czech Academy of Sciences, v.v.i., Husinec-Řež CZ-250 68, Czech Republic. Correspondence and requests for materials should be addressed to T.M. (email: tomas.magna@geology.cz)

Large-scale planetary collisions and smaller impacts are important processes that may add to the chemical makeup of the bodies in the Solar System and shape the planetary surfaces by forming variably sized craters, layers of impact ejecta, and shock deformation features. Some meteorites, in particular carbonaceous chondrites, contain high concentrations of organic matter and volatiles like water, and are considered to potentially bear on the emergence of oceans and life[1]. The identification of projectile types for impacts is thus central for tracing the origin and aftermath of collisions. In general, chondrites are regarded as the most frequent projectiles in larger impacts[2] (see also ref. [3] for the recent summary of meteorite types and individual numbers of specimens). High thermal energy that is released in impacts leads to almost complete melting and/or evaporation of the impactor as well as large amount of the target rocks, and may also change the behavior of the elements and their chemical compounds[4], making the identification of the projectile difficult.

Tektites (distal ejecta) and other impact-related glasses (part of proximal and fallback ejecta) are natural glassy materials which are genetically related to hypervelocity impacts of large extraterrestrial bodies on the Earth's surface. They are produced by intense melting and partial evaporation of the target materials during the impact event, and may bear traces of the impactor[5, 6]. Tektites are found as variably shaped and sized objects; typically, they are of splash-form shapes with a size of few centimetres. They usually are interpreted as being formed from the uppermost layers of the target lithologies, i.e., unconsolidated sediments and soils, partly vaporized and partly melted and ejected with a high velocity from the boundary zone between the impactor and the Earth's surface by jetting[7]. They are mostly silica-rich and with extremely low contents of volatile components ($H_2O$, C, S, halogens, etc.). In fact, tektites belong to the most volatile-depleted natural materials on Earth, often with < 100 p.p.m. $H_2O$[8–10] and low contents of other volatiles (<30–40 p.p.m. C)[11]. There is not an agreement on the exact formation mechanisms of tektites and several concepts have been published[4, 12–14]. A general model of tektite formation and volatile loss[12, 14, 15] developed the original ideas[16] considering possible mechanisms of fragmentation of the overheated tektite melt by separating vapor phase[17, 18]. These ideas were further developed for tektites of the Central European tektite field (moldavites)[4]. This latter conceptual model of formation of tektites considers two main pathways of the behavior of the ejected supercritical fluid during adiabatic decompression: (i) the supercritical matter approaches the sub-critical vapor/melt boundary from the vapor phase stability field and most matter is vaporized, or (ii) the less energetic part of the ejected matter reaches the vapor/melt boundary from the melt stability field and a directly formed overheated melt is fragmented due to separation of the vapor phase (cf. Fig. 1 in refs [17, 18]). The melt fragmentation can strongly be amplified by the escape of water-, carbonate- and/or organic matter-based volatiles derived from the target, which were effectively lost from the tektite melts[9, 11]. Immediate coalescence (accretion) of small melt droplets into larger tektite bodies, probably underscored by formation of late condensation spherules, appears to be the most viable process which may explain the general chemical homogeneity of tektites, in parallel to their large micro-scale chemical heterogeneity, and the general absence of condensation and/or inner diffusion profiles in tektite bodies[4].

Oxygen isotope ($^{16}O$, $^{17}O$, $^{18}O$) ratios have been used to classify asteroidal and planetary materials in the Solar System[19] and O isotopes could thus be used to help identify the impactor. Large mass-independent fractionation (MIF) of oxygen isotopes has been recognized for different Solar System bodies most likely inherited from MIF processes during early Solar System formation, whereas common terrestrial and other planetary materials

(e.g., Mars, Moon, Vesta) have $^{18}O/^{16}O$ and $^{17}O/^{16}O$ ratios which are solely fractionated by mass-dependent processes[20, 21]. The only isotopically anomalous components of Earth are found in the atmosphere. Molecular oxygen has a distinctly negative $\Delta^{17}O$ anomaly[22] (see Methods for definition of $\Delta^{17}O$) that is mainly caused by biological proceses (i.e., photosynthesis, respiration), and by minor effects of photochemical reactions in the upper atmosphere[23]. Due to high temperatures of impact ejecta at the time of formation and during atmospheric re-entry, O isotope anomalies in tektites and related glasses could also be caused by partial post-impact exchange with the Earth's atmosphere[24] although this process has not yet been reported for truly terrestrial samples. In fact, small negative $\Delta^{17}O$ offsets (as low as −0.18‰) were reported for some tektites[20] but it remains unclear whether these low values could provide evidence for such a process or, alternatively, for the presence of impactor material with low $\Delta^{17}O$ values.

The utility of chromium isotope ($^{50}Cr$, $^{52}Cr$, $^{53}Cr$, $^{54}Cr$) systematics to characterize the impactor type was proven in several cases, including the Chicxulub impact at the Cretaceous–Paleogene boundary[2]. Distinct planetary bodies and meteorite groups show resolved differences in the $^{53}Cr/^{52}Cr$ and $^{54}Cr/^{52}Cr$ ratios[25, 26], underscored by particularly high Cr abundance in chondrites, compared to the terrestrial continental crust. Because individual chondrite groups have distinctive $^{17}O$–$^{54}Cr$ isotope fingerprints[3], combined O–Cr isotope measurements may allow identification of impactor material and distinction between impactor- or atmospheric exchange-related variations in $\Delta^{17}O$.

To test the viability of O–Cr isotopes to constrain the impactor type for tektites and impact-related glasses, we measured triple-oxygen isotope compositions for a set of 14 impact-related tektite-like glasses from the Zhamanshin astrobleme (basic splash forms, and acid splash forms—irghizites, two chemically disparate groups of impact-related glasses), and 14 Central European tektites (moldavites), related to the Ries impact event in Germany. Chromium isotope compositions were obtained for two

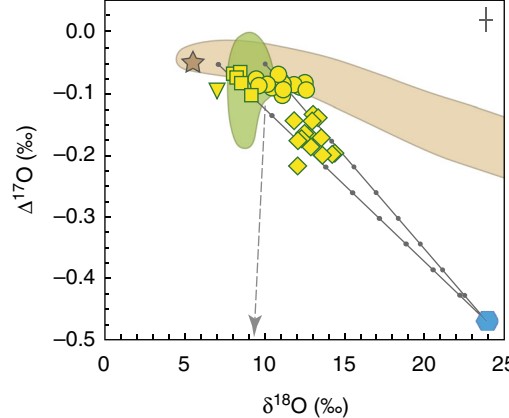

**Fig. 1** Oxygen isotope systematics in tektites and impact glasses from this study. Mixing lines between siliciclastic and mafic materials ($\delta^{18}O$ = 10.0‰ and 7.0‰, respectively) and air $O_2$ are indicated, *bullets* represent 10% increments. *Circles*—moldavites; *diamonds*—irghizites; *squares*—basic splash forms; *reversed triangle*—composite splash-form; *brown field*—the range of common crustal rocks[54] with revised data for San Carlos olivine[76]; *green field*—the recalculated oxygen isotope compositions of other tektites[20] (Australasian tektites, North American tektites). The oxygen isotope compositions of the Earth's mantle (*star*)[76] and air (*hexagon*)[22] are plotted. Carbonaceous chondrite groups other than CI have more negative $\Delta^{17}O$ values (*dashed arrow*). The maximum 2 s.d. error bars for $\delta^{18}O$ and $\Delta^{17}O$ are plotted

impact-related glasses from Zhamanshin, one moldavite, and terrestrial BHVO-2 reference basalt. The detailed analytical procedures for O and Cr isotope measurements and notations for O and Cr isotope compositions are detailed in the Methods section. All samples were previously characterized for major and trace element systematics[4, 27, 28].

Zhamanshin and Ries differ in the size of impactor (estimated diameter of ~0.6 km vs. ~1.2–1.5 km)[27, 29], paleogeography (dry vs. wet area) and indications of the presence vs. absence of extraterrestrial addition in some types of impactites. The Zhamanshin impactor is interpreted to have been about one order of magnitude lower in mass than that of the Ries impact event and it probably was below the size limit necessary for production of distal ejecta, i.e., true tektites. In addition, taking into account the distribution of proximal ejecta, trajectories of both projectiles differed; in case of Zhamanshin the impact must have been much steeper[27] than in case of a relatively shallow trajectory of the Ries projectile[12, 30]. The target surface conditions of Zhamanshin and Ries were also different. While the target of the Ries impact event in the Miocene was a wet area near the limit of the Miocene Upper Freshwater Molasse ('Obere Süsswassermolasse'—OSM) sedimentary basin[31], probably densely overgrown by forests, dry and possibly cold desert conditions are to be expected during the Quaternary in central Asia[27]. Therefore, the quantity of volatiles (derived from $H_2O$, organic carbon, etc.) produced during the Zhamanshin impact can be estimated to be more than one order of magnitude lower than in the case of Ries. The compressed and ejected material of shallow layers of Zhamanshin target thus did not travel to a great distance. Instead, the glass droplets which formed from disintegrated near-surface melts were falling back through the explosion plume and collected small particles both of the mechanically disintegrated impactor matter and condensed particles from the portion of the matter which was evaporated. These glass droplets then coalesced together to form irghizites[27, 28]. In a number of cases, the surface layers of primary droplets within irghizites are strongly enriched in Fe, Mg, Cr, Co, Ni and P[28, 32].

We explore several types of glasses from the Zhamanshin impact structure (48°24′N/60°58′E), all belonging to the fallback and proximal ejecta. The major focus of the study is on tektite-like splash-form glassy objects sized usually up to several centimetres. Three major chemical types are distinguished: (i) $SiO_2$-rich (usually 69–76 wt.% $SiO_2$) irghizites[33, 34] which have commonly been formed by coalescence of < 1 mm glass droplets and have elevated concentrations of Ni, Co and Cr[27, 28], (ii) morphologically rather uniform drops and their fragments composed of more basic glass, termed 'basic splash forms' (53–56 wt.% $SiO_2$)[27, 28, 32, 35–38], and (iii) composite splash forms, defined as inhomogeneous acidic splash forms with abundant mineral inclusions[28]. Impact glasses occurring in larger irregularly shaped fragments and large blocks called 'zhamanshinites'[33] were not the subject of this study because earlier studies excluded any presence of the impactor matter in these objects[28]. We refer to the Supplementary Note 1 for further details.

Tektites of the Central European strewn field (moldavites) are genetically related to the Ries Impact Structure, Germany (48° 53′N/10°33′E), a complex type astrobleme with a diameter of ~26 km[39]. Unconsolidated sandy, silty and clayey sediments of the OSM, probably covering most of the target area at the time of the impact, are generally considered to be the dominant source material of moldavites[4, 13, 29, 30, 39–46]. The origin of moldavites in relation to the Ries Impact Structure has also been confirmed by generally consistent ages of moldavites and impact-related glasses from the Ries Impact Structure itself at $14.75 \pm 0.20$ Ma[31, 47]. The nature of the impactor remains elusive[48, 49] due to difficulties in finding unequivocal extraterrestrial chemical signatures both in moldavites and impact-related glasses[50].

**Table 1 Oxygen isotope composition of impact-related materials from this study**

| | $\delta^{18}O$ (‰)[a] | $\delta^{17}O$ (‰) | $\Delta^{17}O$ (‰)[a] | n |
|---|---|---|---|---|
| *Tektites* | | | | |
| SBM-11 | 11.18 | 5.83 | −0.088 | 1 |
| SBM-11 replicate[b] | 9.62 | 5.02 | −0.090 | 1 |
| SBM-23 | 10.98 | 5.72 | −0.080 | 1 |
| SBM-35 | 12.46 | 6.50 | −0.084 | 2 |
| SBM-44 | 10.95 | 5.71 | −0.086 | 2 |
| SBM-88 | 12.57 | 6.55 | −0.096 | 3 |
| SBM-192 | 11.79 | 6.15 | −0.085 | 1 |
| MM-60 | 10.85 | 5.65 | −0.092 | 2 |
| MM-67 | 10.84 | 5.66 | −0.070 | 1 |
| CHBM-5 | 11.81 | 6.15 | −0.086 | 2 |
| CHBM-6 | 9.44 | 4.92 | −0.078 | 1 |
| CHBM-7[b] | 10.13 | 5.30 | −0.086 | 2 |
| MCB-2 | 12.14 | 6.33 | −0.091 | 2 |
| LM-1 | 11.10 | 5.79 | −0.085 | 2 |
| MOLD-SB[b] | 11.08 | 5.30 | −0.105 | 1 |
| MOLD-SB replicate[b] | 10.45 | 5.79 | −0.092 | 1 |
| MOLD-SB replicate[b] | 11.12 | 5.47 | −0.095 | 1 |
| *Irghizites* | | | | |
| IRG-IZ1 | 12.76 | 6.58 | −0.170 | 2 |
| IRG-IZ3 | 11.85 | 6.12 | −0.145 | 2 |
| IRG-IZ3 replicate | 13.36 | 6.92 | −0.141 | 1 |
| IRG-IZ3 replicate | 13.05 | 6.76 | −0.135 | 1 |
| IRG-IZ3 replicate[b] | 12.98 | 6.76 | −0.147 | 1 |
| IRG-IZ9 | 14.42 | 7.42 | −0.198 | 2 |
| IRG-IZ9 replicate[b] | 13.59 | 7.03 | −0.201 | 1 |
| IRG-IZ9 replicate[b] | 13.49 | 7.01 | −0.173 | 1 |
| IR-2 | 14.24 | 7.33 | −0.200 | 1 |
| IR-4 | 12.30 | 6.33 | −0.175 | 1 |
| IR-8 | 12.49 | 6.44 | −0.166 | 1 |
| IR-9 | 12.09 | 6.18 | −0.219 | 1 |
| IR-11 | 12.90 | 6.63 | −0.188 | 1 |
| IR-12 | 12.10 | 6.22 | −0.178 | 1 |
| *Basic splash forms* | | | | |
| IRG-M4 | 8.01 | 4.17 | −0.068 | 2 |
| IRG-M4 replicate | 9.18 | 4.76 | −0.104 | 1 |
| IR-10 | 8.47 | 4.42 | −0.066 | 1 |
| IR-13 | 8.26 | 4.30 | −0.075 | 1 |
| IR-14 | 8.56 | 4.45 | −0.084 | 1 |
| *Composite splash forms* | | | | |
| IR-7 | 7.03 | 3.63 | −0.094 | 1 |

[a]Estimated measurement uncertainties are <0.3‰ and <0.008‰ for $\delta^{18}O$ and $\Delta^{17}O$, respectively, based on the typical reproducibility (1 s.d.) of results for the standard reference material NBS 28 throughout the analytical sessions, except for samples denoted with superscript 'b'
[b]Samples analyzed with an improved extraction line (see text for details). Estimated measurement uncertainties are <0.15‰ and <0.007‰ for $\delta^{18}O$ and $\Delta^{17}O$, respectively, based on the typical reproducibility (1 s.d.) of measurement results for San Carlos olivine

## Results

**Oxygen isotopes.** The $\Delta^{17}O$ values of moldavites (from −0.11 to −0.07‰; Table 1) overlap with crustal rocks (Fig. 1) and the range of $\delta^{18}O$ values from 9.4 to 12.6‰ exceeds that reported previously (from 10.7 to 11.9‰)[41, 51–53]. The larger variability in $\delta^{18}O$ likely is a consequence of analyzing a chemically more variable suite of moldavites[4, 29]. The new data set confirms the ~3–5‰ offset in $\delta^{18}O$ between moldavites and unconsolidated OSM quartz-rich sands (Fig. 2), where the likely major moldavite source material analogs such as bulk Miocene sediments from southern surroundings of the Ries Impact Structure showed distinctly higher and more variable $\delta^{18}O_{SMOW}$ values from 14.1 to 23.3‰ (ref. [41]). The OSM sands, which are are assumed to be the dominant source for moldavites[4, 41], showed $\delta^{18}O$ values from 14.1 to 16.5‰ with an average of 15.3‰ (ref. [41]) and the predominance of sands in moldavites is also reflected in low MgO and CaO contents and moderate $\delta^{18}O$ (Fig. 3), excluding large

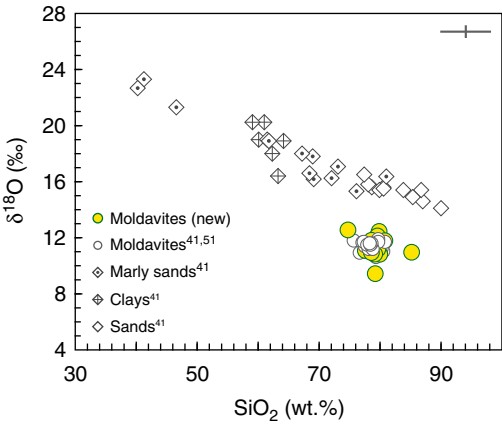

**Fig. 2** New and published $\delta^{18}O$ values vs. $SiO_2$ contents in moldavites and sediments from the Ries area. The wider range in $\delta^{18}O$ values of moldavites from this study is due to analyzing chemically more variable specimens[29] compared to those analyzed earlier[41, 51]. Silica contents in some earlier analyzed moldavites[51] were reported elsewhere[83]. Available combined $SiO_2$-$\delta^{18}O$ data for chemically diverse Ries area sediments[41] are plotted. Clays and marly sands can represent only a small proportion of parental material to moldavites while $SiO_2$-rich sands dominate the budget[4, 13, 41]. The consistent $\delta^{18}O$ offset of ~3–5‰ between moldavites and possible parental sediments from Ries area appears to be a combination of several aspects, explored in detail in the main text. The maximum 2 s.d. error bars for $\delta^{18}O$ (±0.4‰) and $SiO_2$ (±2%) are plotted

volumes of carbonates in moldavite melts. The basic splash forms from Zhamanshin show $\Delta^{17}O$ values (−0.10 to −0.07‰) that also overlap with crustal rocks. The $\Delta^{17}O$ values measured for irghizites (−0.22 ⩽ $\Delta^{17}O$ ⩽ −0.14‰) are significantly lower than those measured for basic splash forms, for the moldavites, and also those reported for common crustal rocks (ref. [54] and discussion therein; Fig. 1). This difference in underscored by distinct $\delta^{18}O$ values for basic splash forms and irghizites (8.0–9.2‰ vs. 11.9–14.4‰). The $\Delta^{17}O$ = −0.09‰ for composite splash-form IR-7 is at the low end of values for a given $\delta^{18}O$ = 7.0‰ (Fig. 1).

**Chromium isotopes**. Irghizite IR-8 and composite splash-form IR-7 display a significant positive anomaly in $^{54}Cr$ ($\varepsilon^{54}Cr$ up to 1.54) which is clearly outside the terrestrial range (Fig. 4) and indicates extraterrestrial origin of a large part of $Cr^{27, 28}$ in IR-8 and IR-7. This is different from the results for moldavite SBM-88 and basalt BHVO-2 (Table 2), which are indistinguishable from typical terrestrial values[26, 55, 56].

## Discussion

Moldavites have uniform terrestrial $\Delta^{17}O$ values (mean $\Delta^{17}O$ = −0.09 ± 0.02‰, 2 s.d.) that are consistent with common sedimentary precursors and weathered residues[4, 41] (Fig. 1). This indicates insignificant contamination by isotopically anomalous extraterrestrial matter. Neither is any exchange with air $O_2$ obvious. The absence of any apparent evidence for impactor material is underscored by Cr isotope data for the slightly Cr-enriched[57] moldavite SBM-88 with $\varepsilon^{54}Cr$ = 0.09 ± 0.07 (2 s.e.), which is identical to the values reported for a range of pristine terrestrial materials[56] (Table 2). Because Cr is a sensitive tracer of chondritic contamination in impact-related materials[2, 58], the new coupled $^{17}O$–$^{54}Cr$ isotope data support the lack of extraterrestrial addition for moldavites outside the currently achievable analytical precision for these elements.

The consistent negative $\delta^{18}O$ offset of ~3–5‰ found for moldavites relative to the Ries area target rocks is extended for a

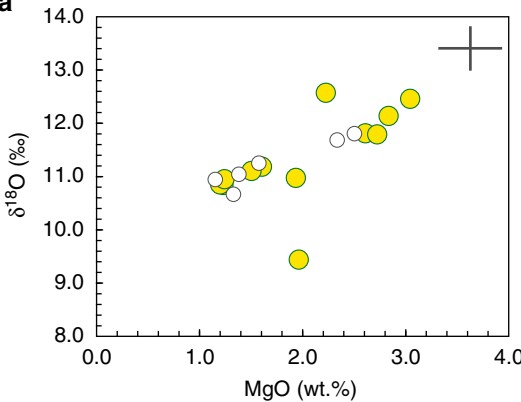

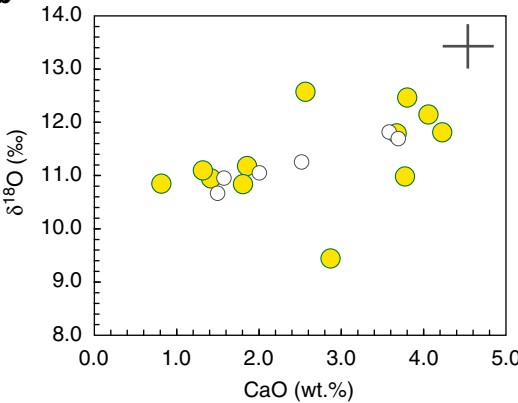

**Fig. 3** New and published $\delta^{18}O$ values vs. MgO **a** and CaO **b** in moldavites. The wider range in $\delta^{18}O$ values from this study (*yellow circles*) is not reflected in distinctive major element compositions[4], compared with other published data[51, 83]. The 2 s.d. error bars are plotted

larger chemical range of moldavites, analyzed in this study (Fig. 2; this study and refs. [41, 51]), and requires mass-dependent $^{18}O$ depletion (Fig. 1). This offset can generally be interpreted by either addition of a low-$\delta^{18}O$ component, or additional mass-dependent O isotope fractionation during tektite formation[41, 59]. To explain this several per mil offset, incorporation and isotope homogenization of ~22 vol.% of a low-$\delta^{18}O$ water (−10‰) partly filling ~40–45 vol.% of pore spaces of the OSM sands, followed by conversion of the matter to plasma and its condensation back to a silicate glass was advocated[13, 41]. However, to shift the $\delta^{18}O_{SMOW}$ of the moldavite parent mixture from the OSM sand average value of 15.3‰ to the moldavite average value of 11.5‰ would require addition (and full isotope homogenization) of a much higher proportion of meteoric water. The mid-Miocene climate of this part of Europe was significantly warmer than modern climate[60, 61] and the average $\delta^{18}O$ values of Miocene meteoric water were also higher, most probably in the range −5.6 ± 0.7‰ (ref. [62]). This higher $\delta^{18}O$ value would further increase the necessary proportion of fully homogenized meteoric water to unrealistic levels at ~30–40 vol.% of the mixture.

The Fig. 2 clearly shows that the carbonate-rich OSM sediment types could not have been an important component in the moldavite parent mixture. The OSM sands and clayey sands, with only a minor carbonate component, are much more suitable with respect to both major element chemistry and O isotope composition. Correlations between the major element contents and $\delta^{18}O$ are generally poor (Fig. 3). The dolomitic carbonate, present as a minor clastic component in some types of OSM sands[4], was clearly present but only at quantities of a few per cent, at

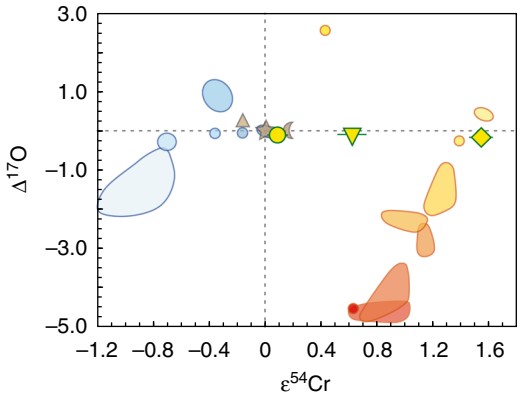

**Fig. 4** $\Delta^{17}O$ vs. $\varepsilon^{54}Cr$ in impact glasses and tektites from this study compared with chondrites, achondrites and large planetary bodies. The plot is modified from refs. [3, 26, 58] with new data from this study plotted as *yellow symbols* (*circle*—SBM-88; *reversed triangle*—IR-7; *diamond*—IR-8), with 2 s.e. uncertainties plotted. For oxygen, the error bars are smaller than the corresponding symbol size. Different hues of orange from light to dark represent different classes of carbonaceous chondrites (CI, Tagish Lake, CR, CB, CM, CV, CO, CK—*red spot*), the single point with high $\Delta^{17}O$ ~2.5‰ represents Rumuruti chondrites. Different hues of blue from light to dark represent ureilites (farthest from Earth), mesosiderites–HED meteorites–pallasites–IIIAB irons group, ordinary chondrites, aubrites, angrites and enstatite chondrites (near-identical to Earth). *Triangle*—Mars; *star*—Earth; *crescent*—Moon

**Table 2 Chromium isotope systematics of impact-related materials from this study**

|  | Cr (p.p.m.) | $\varepsilon^{53}Cr$ | 2 s.e. | $\varepsilon^{54}Cr$ | 2 s.e. | n |
|---|---|---|---|---|---|---|
| *Irghizites and splash forms* |  |  |  |  |  |  |
| IR-8 | 213[a] | 0.31 | 0.03 | 1.54 | 0.08 | 4 |
| IR-7 | 122[a] | 0.15 | 0.05 | 0.62 | 0.10 | 2 |
| *Tektites* |  |  |  |  |  |  |
| SBM-88 | 50[b] | 0.01 | 0.04 | 0.09 | 0.07 | 4 |
| *Terrestrial reference materials* |  |  |  |  |  |  |
| BHVO-2 | 280[c] | 0.02 | 0.03 | 0.07 | 0.07 | 2 |

The letter *n* represents the number of individual runs where each single run consists of four successive measurements in static mode with isotopes shifted by one mass unit in the Faraday detectors
[a]Data from ref. [28]
[b]Data from ref. [57]
[c]Certified value from US Geological Survey

maximum[63]. Such a small admixture of high-$\delta^{18}O$ carbonate was thus only one of several factors controlling the $\delta^{18}O$ range of moldavites. The carbonate as an important source component is also ruled out from high $^{87}Sr/^{86}Sr$ ratios in moldavites[42, 64], which indicate that a major part of Sr in moldavites is derived from silicates rather than from clastic carbonates. Similarly, the documented mass-independent effects[65] during high-temperature decomposition of carbonates would have only a limited effect on $\Delta^{17}O$ of moldavites as evidenced by the measured values which are not shifted from the common crustal range.

Therefore, it can be assumed that the observed $\delta^{18}O$ values of moldavites resulted from a combination of several factors: (i) mixing of quartz, clay, carbonate and some other minor components in varying proportions; (ii) addition of ~10–30 vol.% of water into the parent mixture; and (iii) isotope fractionation between the molten glass and separated volatiles at temperatures above 1200 °C. These findings are also supported by moderate to significant losses of highly and moderately volatile elements[4, 13, 41, 66–68], while less volatile elements appear to have been quantitatively transported into moldavite melts without accompanying isotope fractionation[57, 63, 69]. Unfortunately, the O isotope fractionations for the gas phase-melt systems are not calibrated in the temperature range of interest (i.e., between the melting temperature of moldavite glass and that of lechatelierite, generally above 1200 °C). By projection of the available fractionations[70] into this high-temperature region it can be assumed that escaping volatiles (mainly CO, $CO_2$) should have their $\delta^{18}O$ values several tenths or a few per mil units higher than the silicate melt, shifting the residual melt $\delta^{18}O$ to a lower values.

In contrast to a resolved difference in $\delta^{18}O$ between moldavites and their inferred parent materials (this study and refs. [41, 51]), overlapping O isotope compositions were found for the Ivory Coast tektites, and their likely parental metasedimentary rocks and granitic dykes in the Bosumtwi impact structure, Ghana[71]. Therefore, the magnitude of the $\delta^{18}O$ shift observed during tektite formation can perhaps be related to the quantity of volatile-bearing species (water, carbonates and organic matter)

present in the target area. Such an observation implies that solely the transformation of target materials into tektites does not generate mass-independent effects and other pertinent processes are required to modify $\Delta^{17}O$.

Unlike the constrained crustal-like $\Delta^{17}O$ values reported for moldavites, oxygen isotope systematics of irghizites and splash forms from Zhamanshin show several peculiar traits. The most appealing feature of the data set are the uniformly low $\Delta^{17}O$ values in the irghizites ($\leqslant -0.14‰$) relative to basic splash forms ($\geqslant -0.10‰$) while, at the same time, irghizites display higher $\delta^{18}O$ values (Fig. 1). The difference in chemical composition between irghizites and basic splash forms reflects differences in the composition of target materials in Zhamanshin area[28]. Basic splash forms sampled deeper-seated lithologies such as andesitic volcanic rocks and basement tuffs, while irghizites were formed from surface sands and clays[27, 28, 72], evidenced also by distinct Sr–Nd isotope systematics[64] and cosmogenic $^{10}Be$ identified solely in the irghizites, which suggests sampling of surface material[73]. This provides unequivocal evidence for the depth segregation limiting the interaction between impactor and target rocks for Zhamanshin impact materials but does not explain the difference in $\Delta^{17}O$ between basic splash forms and irghizites (Fig. 1). The observation is discussed in terms of two different scenarios: the anomaly is due to admixture of anomalous oxygen from the extraterrestrial projectile, or the impact melts may have exchanged with isotopically anomalous air oxygen in the aftermath of the impact. We assume a $\delta^{18}O \approx 10‰$ for the end member target rock of irghizites, which is compatible with their sedimentary siliciclastic source[74].

The low $\Delta^{17}O$ in irghizites could be caused by contamination with anomalous (i.e., $\Delta^{17}O < -0.5‰$) oxygen from the projectile. While both chondritic and iron meteorites were proposed as possible impactors for Zhamanshin[27, 28, 36], in the context of an impactor origin of the O isotope anomaly in irghizites, ordinary chondrites and iron meteorites would be excluded as impactor. Addition of ~2–10% oxygen from a carbonaceous chondrite impactor would explain the O isotope systematics of irghizites but most classes of carbonaceous chondrites were excluded on the basis of siderophile element systematics[28]. The abundances of highly siderophile (HSE—Os, Ir, Ru, Pt, Pd, Re) and moderately siderophile (MSE—Ni, Co, Cr) elements as an indicator for impactor contribution differ greatly between irghizites and basic splash-forms[28]. In particular, Ni, Co and Cr are enriched in irghizites relative to basic splash forms (up to ~2000 p.p.m. Ni in the former)[27, 28, 36], suggesting a resolved addition from a Ni-rich chondrite. To qualify the impactor more precisely, the unique $^{54}Cr$ excess of 1.54ε units found for irghizite IR-8 (Table 2 and

Fig. 4) represents the highest anomaly ever observed in any terrestrial sample, including those with resolved meteoritic addition[58], and among the highest anomalies ever reported for any Solar System material[26, 56]. This high $\varepsilon^{54}$Cr falls in the field intermediate between CI-type chondrites and the ungrouped Tagish Lake chondrite. The observed Cr isotope composition clearly is inherited from the impactor and disqualifies most carbonaceous chondrites as an impactor for the Zhamanshin astrobleme. A simple model (Fig. 5) demonstrates that $^{54}$Cr systematics in Zhamanshin impact glasses can be reproduced by mixing with 2–7% of CI chondrite having 2650 p.p.m. Cr and $\varepsilon^{54}$Cr = 1.7 (refs. [56, 75]). It also requires two different target lithologies with low (IR-8) and intermediate (IR-7) Cr abundance, fully consistent with recent investigations[27, 28] and oxygen isotope constraints (this study). At present, combined $^{54}$Cr–$^{17}$O isotope data are only collected for a limited number of individual carbonaceous chondrites and the possibility of the existence of chondrites with yet unaccounted $^{54}$Cr–$^{17}$O isotope systematics is not eliminated. For example, ~5% admixture of a hypothetical Ni-rich carbonaceous chondrite with $\varepsilon^{54}$Cr = 1.6 and $\Delta^{17}$O of ~ −2‰, which would fall close to the field of CR carbonaceous chondrites, would explain the observed composition of irghizites without a need for isotope exchange with Earth's atmosphere, discussed below. However, a meteorite with such an isotope signature has not yet been reported and $^{54}$Cr isotope signature in particular represents a serious constraint for tracing a suitable impactor type.

The Earth's atmospheric molecular oxygen carries a $\Delta^{17}$O value of −0.47‰ (ref. [22]); with a correction of $\delta^{17}$O (see ref. [76] for discussion). At high temperatures, a limited extent of mass-dependent isotope fractionation is expected between molecular $O_2$ and silicate melt so that the composition of the irghizites may represent a mixture of the target rocks and air $O_2$ (Fig. 1). In such a scenario, 20–35% of the oxygen in the irghizites would have equilibrated with air $O_2$ while it would be generally < 10% for

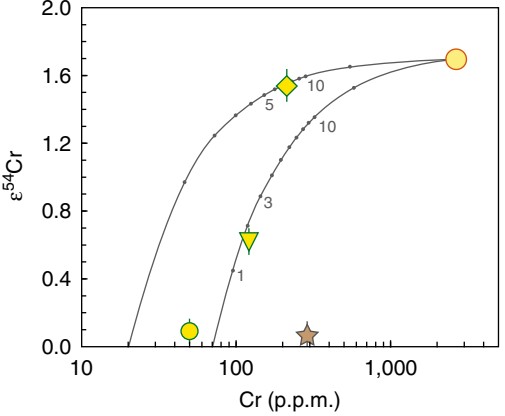

**Fig. 5** Simple binary mixing model for impact glasses from Zhamanshin with CI chondrite impactor. CI chondrite with 2650 p.p.m. Cr and $\varepsilon^{54}$Cr = 1.7 (*yellow circle* with a *red rim*; see main text for data sources) was mixed with different target materials, a Cr-poor (20 p.p.m.) composition for irghizite IR-8 (*diamond*) and intermediate-Cr (70 p.p.m.) composition for composite splash-form IR-7 (*reversed triangle*), reflecting their different sources[28]. The numbers along the calculated mixing lines indicate a proportion of the impactor matter in the mixture. The $^{54}$Cr results are identical to $^{53}$Cr results (not shown), supporting the robust nature of the mixing process. The nil effects for moldavite SBM-88 (*yellow circle* with a *green rim*) and terrestrial standard BHVO-2 (*star*) are documented. The errors for $\varepsilon^{54}$Cr values are 2 s.e. (see Table 2), the errors for Cr concentrations (1–3% RSD)[28] are smaller than the corresponding symbol size

basic splash forms. This scenario would account for the low $\Delta^{17}$O in irghizites compared to basic splash forms. It also supports the recent models of formation of impact glasses from Zhamanshin[27, 28], in which small particle-sized irghizites represent the last phase of fallback deposition, thus allowing for the extended period of time in contact with a progressively cooling and collapsing vapor cloud, in comparison with larger particles such as basic splash-forms[77]. A detailed petrographic and microchemical study confirmed that mm- to cm-sized irghizite bodies frequently formed by coalescence of ~1 mm large glass droplets, a feature not observed for basic splash forms[28]. A surface layer 0.1 mm thick represents ~50% of the volume of such a sphere. Iron in interior parts of these glass spheres is in $Fe^{2+}$ form whereas more oxidized iron is found in surface layers. This is indicated by the observation that rims of the spheres display lower analytical totals and higher total Fe contents than their interiors. In particular, the analytical totals for the rims vary between 97 and 99 wt.% and $FeO^{tot}$ ranges from 8.6 to 10.4 wt.%, while the inner parts of the spheres show analytical totals of ~100 wt.% with $FeO^{tot}$ ranging from 5.6 to 7.0 wt.%. The balancing of the analytical totals in rims requires some 30–50% of total iron in rims to be present as $Fe^{3+}$. Apparently, lower analytical totals associated with Fe enrichment in the rims suggest the origin of rims in a more oxidizing environment than for interiors of spherules, indicating exchange with the atmosphere[24]. We note that the measured $\Delta^{17}$O data are from bulk homogenized irghizite samples and that in-situ $\Delta^{17}$O values would likely vary strongly within individual irghizite bodies. The model of slow gravitational settling[77] and chemical and isotope exchange with a slowly collapsing vapor plume cannot be applied to moldavites (distal ejecta) due to their instantaneous dislocation from the area of origin.

Combining O and Cr isotope data allow both classifying the impactor type and pinpointing the process, which gives rise to the unusually low $\Delta^{17}$O in irghizites. The $\Delta^{17}$O of both CI carbonaceous chondrites and Tagish Lake chondrite is close to the terrestrial fractionation line. The Cr concentration in irghizite IR-8 is 213 p.p.m., whereas it is presumably low (~20 p.p.m.) in possible target lithologies[28]. The $^{54}$Cr systematics indicates that >90% of Cr is derived from the impactor (Fig. 5), and the general $^{54}$Cr systematics disqualifies most types of carbonaceous chondrites (Fig. 4). Mixing a target material ($\delta^{18}$O = 10‰; $\Delta^{17}$O = −0.09‰) with a CI-like impactor having $\delta^{18}$O = 16.3‰ and $\Delta^{17}$O = 0.2 ± 0.2‰ (ref. [78]) (Tagish Lake chondrite has a similar composition) would not result in the observed low $\Delta^{17}$O of the irghizites. Therefore, we conclude that exchange with atmospheric $O_2$ in a vapor plume could lead to the observed low $\Delta^{17}$O in irghizites. Apart from rare sulphates[79] and fossil biominerals[80, 81], this makes irghizites of the Zhamanshin astrobleme one of the few terrestrial materials in the geological record that carries traces of the isotope anomaly of air $O_2$.

## Methods
**Oxygen purification and isotope measurements**. The procedures for the oxygen extraction and high-precision triple-oxygen isotope measurements followed the methodology outlined elsewhere[54]. In brief, ~2 mg of a powdered sample were reacted with fluorine gas at a low pressure (~20 mbar) and heated with a $CO_2$-based laser. Excess fluorine was removed by hot NaCl and chlorine was subsequently trapped with liquid nitrogen cold trap. Oxygen liberated from analyzed samples was collected on 5 Å molecular sieves and cleaned by gas chromatography for mass spectrometry analysis. For some samples, oxygen was extracted using an improved protocol that applies $BrF_5$ as the fluorinating agent (details of this protocol are given elsewhere[76]). All three oxygen isotopes were measured using a Finnigan MAT 253 gas source mass spectrometer, housed at the Göttingen University. The results are reported in $\delta^{17}$O–$\delta^{18}$O (‰) notation and $\Delta^{17}$O (‰) values relative to VSMOW2 reference material. The $\Delta^{17}$O is defined relative to a reference line with the slope of 0.5305 and zero intercept[54] as $\Delta^{17}$O = 1000ln($\delta^{17}$O/1000 + 1) − 0.5305 × 1000ln($\delta^{18}$O/1000 + 1), with San Carlos olivine having a $\Delta^{17}$O = −0.05‰ (ref. [76]). The 1 s.d. reproducibility of the measurements was always better than 0.3‰ for $\delta^{18}$O and 0.01‰ for $\Delta^{17}$O values, respectively, based on repeated

analyses of reference quartz NBS-28 (National Bureau of Standards, USA) and San Carlos olivine.

**Chromium purification and isotope measurements**. The procedures for the Cr isotope measurements were performed at the Institut de Physique du Globe de Paris, France. The two irghizite samples (IR-7 and IR-8), one moldavite sample (SBM-88) as well as the basalt BHVO-2 (Hawaii, USA) were dissolved in a mixture of concentrated HF and $HNO_3$ in Teflon bombs at 140 °C for several days until complete dissolution of all phases. Chromium was purified by cation exchange chromatography following the method described elsewhere[55]. The $^{53}Cr/^{52}Cr$ and $^{54}Cr/^{52}Cr$ isotope ratios were measured using a Triton thermal ionization mass spectrometer (Fisher Scientific, Bremen, Germany) following the method described elsewhere[26]. Briefly, samples were loaded in chloride form on the outgased W filaments together with an Al-silicagel–$H_3BO_3$ emitter. Each run comprised 20 blocks of 20 cycles and each single run was a combination of three successive multi-collection measurements in static mode. The measured $^{53}Cr/^{52}Cr$ and $^{54}Cr/^{52}Cr$ ratios are reported in $\varepsilon^{53}Cr$ and $\varepsilon^{54}Cr$ notation (1 in 10,000 difference relative to NIST SRM 3112a reference material) and were normalized using an exponential law to $^{52}Cr/^{50}Cr = 19.28323$ (ref. [82]). Each sample was measured two to four times (Table 2).

**Data availability**. All relevant data are available from the authors on request and/or are included with the manuscript in the form of data tables and data within figures.

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

## Acknowledgements

This study was supported by the Czech Science Foundation project 13-22351S. F.M. thanks the ERC under the European Community's H2020 framework program/ERC grant agreement #637503 (Pristine), the ANR for a chaire d'Excellence Sorbonne Paris Cité (IDEX13C445) and the UnivEarthS Labex program (ANR-10-LABX-0023 and ANR-11-IDEX-0005-02). Part of this work was supported by IPGP multidisciplinary program PARI, and by Region île-de-France SESAME Grant no. 12015908.

## Author contributions

T.M. and K.Ž. conceived the study. R.S., J.M. and Z.Ř. provided the samples. A.P., S.P., F. and B.M. collected the data. Š.J. and R.S. characterized the microchemistry of key samples. All authors contributed to data interpretation. T.M., K.Ž., A.P. and F.M. took lead in writing.

## Additional information

**Competing interests:** The authors declare no competing financial interests.

