## [Peer Review File · Nature Communications]

Reviewers' Comments:

Reviewer #1 (Remarks to the Author)

The Zhamanshin impact glass has been demonstrated to contain a significant amount of extraterrestrial material (chondritic material in the 1-10% range) based on Ni, Co, Cr contents (Jonasova et al. 2016). These authors suggested a carbonaceous chondrite impactor possibly CB-CH type due to the high Ni/Cr ratio.

The present paper by the same group proposes to further constrain this parent body classification using two isotopic systems, oxygen and chromium. In both systems an ET signature seems demonstrated, obviously with Cr, likely with O as the alternative of an air contamination is difficult to account for. Nevertheless the shift in D17O is minor: -0.1‰, i.e. in the range of crustal rocks. Both isotopic results point toward a carbonaceous chondrite like impactor, confirming Jonasova et al. paper. This is worth publishing, possibly in a more specialized journal as these results do not lead to deeply innovative conclusions.

However, the way the isotopic couple is used to propose a Tagish lake like impactor denotes a major misunderstanding of what two component mixtures produce on isotopic systems.

In particular the idea (implicit in the paper and Fig.3) that in a two isotopic diagram the mixture should plot on a straight line between the two end-members is plain wrong.

Let's precise the problem:

Trace elements have set an upper limit of ET contamination of 10% in mass. As a chondrite have an oxygen content only slightly less than a standard crustal rock, the amount of ET oxygen in the irghizite is less than 10%, say 5%. The 0.1 shift in thus translates into a D17O of the impactor below -2 ‰. This definitely excludes a Tagish lake like parentage and points toward CB or other carbonaceous chondrite parentage.

Concerning the Cr content, Jonasova results imply that a large majority of Cr in irghizite is of ET origin, thus the $\epsilon^{54}\text{Cr}$ will be close to the parent body value. The 1.5 value seems to point toward TL, but the possibility that it could fit with other CC groups is dismissed by the wrong visual impression that both oxygen and Cr values should fit the parent body!

What is also missing is a discussion of the variability of $\epsilon^{54}\text{Cr}$ in CC and the possible effect of impact volatilization on $\epsilon^{54}\text{Cr}$. It would have been clever to obtain data on moldavite to assess this effect as the present paper propose moldavite as a clear exemple of impact glass devoid of ET contamination.

A further example of major reasoning flaws can be found in the following sentence:

"A tight correlation of $\epsilon^{54}\text{Cr}$ versus $\Delta^{17}\text{O}$ for irghizite » where is this correlation? When one has only two data points how can one put forward such a claim ?!

Reviewer #2 (Remarks to the Author)

Issaku Kohl: review of NCOMMS-16-20902-T

Zhamanshin astrobleme: O-Cr isotope evidence for a carbonaceous chondrite impactor

Summary of Approach

The authors suggest that O and Cr (although much less Cr data is presented) isotope fingerprinting can identify impactor assimilation in tektite and similar type impact glasses.

Oxygen and chromium isotopic compositions of extraterrestrial materials have been extensively characterized and expected ranges for different impactor types are previously documented. Given well-characterized target rock lithologies and isotopic compositions, this is a valid approach.

This approach is applied to impact glasses thought to be associated with 2 impact structures, Zhamanshin (irghizites and more basic splash forms) and Ries (moldavites). The data shows significantly negative $\Delta^{17}\text{O}$ (twice as negative as terrestrial rocks) values for irghizites, while the basic splashforms from Zhamanshin and the moldavites show nominally terrestrial values.

General Comments and Suggested Changes

The authors effectively discount atmospheric oxygen assimilation as a possible source for the $\Delta^{17}\text{O}$ anomaly as it requires incorporation of an unreasonable amount of $\Delta^{17}\text{O}$ negative and $\delta^{18}\text{O}$ enriched air O_2 (-0.475‰ and 23.8‰ respectively). They also suggested that the terrestrial $\Delta^{17}\text{O}$ bearing basic splash forms and moldavites are simply lacking in impactor assimilation, appealing to the two rock types sampled by the Zhamanshin impact having different sampling depths. While not explicitly stated I believe the authors are appealing to depth segregation limiting interaction between impactor and basic target rocks for Zhamanshin. If that is the case, it should be stated explicitly. As it stands, there is no good explanation for acidic tektites bearing impactor related anomalies and basic ones not, this is a crucial observation that must be addressed

A brief discussion of siderophile element content and Cr isotope composition in irghizites allow the authors to discount major meteorite classes as possible impactors settling on a Tagish Lake type ungrouped chondrite as the most likely candidate. This needs to be highlighted in a more aggressive way. I believe this to be the most significant contribution in this manuscript. Currently, it reads as though the non-anomaly bearing tektites are the minority and much time and effort is put toward explaining why they have no anomaly. However, given that 2 out of 3 types of tektites (the majority) in this manuscript have no anomaly it seems pertinent to start with that as the "easy to explain case" and then go into the "special case" where an anomaly is inherited. This adds weight and sets the reader up for the ungrouped chondrite conclusion, which is the major contribution.

In addition, I believe a short discussion on evaporation is needed. The authors allude to it in the last paragraph with one reference. For "impact folks" the formation of evaporative vapor condensates is a given and we know that the isotope effects will differ for vapor and liquid. I am not suggesting that this process plays a major role but I would like to see it receive some treatment. A few calculations showing a more quantitative attempt to model the potential role of evaporation, even in the supplement, would add to the thoroughness.

Recommendation

Aside from the above suggestions, the manuscript is well written and the data look excellent. As to reproducibility of the data set, there are only a few labs in the world that are doing $\Delta^{17}\text{O}$ at the level of the Pack group at Gottingen, so for those other groups it should be attainable. All that said, I believe this is a good approach to identifying elusive impactor isotopic and chemical compositions from preserved impact glass sediments. It will benefit from some reorganization and a bit of additional information but I recommend this for publication with significant changes.

Best,

Issaku Kohl Ph.D.

Project Scientist II
Young Stable Isotope Laboratory
595 Charles E. Young Drive East
2676 Geology Building
Los Angeles, CA, 9095
ikohl@epss.ucla.edu

Reviewer #3 (Remarks to the Author)

Please see text, table, and figure in the uploaded DOC file.

Thanks to the editor for the opportunity to review this most interesting submission to Nature Communications. The work is of particular interest because it reports data strongly reminiscent of samples taken from the rain gutters of houses in the Chicago suburb of Park Forest shortly after the fall of a large meteorite on 26 March 2003. The samples were collected by meteorite collectors and sent to me for O-isotope analysis by Tony Irving (Univ. Washington). I'm not writing a conventional ms. review because I think data from Park Forest may help the authors of the ms. to reach a better interpretation of the origin of the aerodynamically-shaped glassy forms found at Zhamanshin.

When I received the samples from Tony Irving, I was astonished to see, under a binocular microscope, glassy mm-sized aerodynamic forms. Dumbbell shapes were common with two bulbous ends separated by a narrow connecting filament. There were also half-dumbbells with a single bulbous end and a filament tapering to a sharp end. As I recall, there was a single example of a specimen that had its filament folded back upon itself giving the shape of a jug with a curved handle. I can't send photos because I'm in Paris, giving a seminar at I'IPGP on Friday, and can't access records stored in my lab in Washington, DC.

Above, I have copied analytical data on the O-isotopes of the dumbbells. In the data table, duplicate analyses of the same gas sample are given for run 12-4. The other runs, 11-204 and 12-9 are single analyses of samples fluorinated at different times. I used BrF₅ as fluorinating reagent and heated the samples to promote fluorination with a CO₂ laser and calculated $\Delta^{17}\text{O}$ with the factor 0.526. As can be seen in the data, the Park Forest dumbbells are remarkably similar to the O-isotope data of the irghizites: In the order $\Delta^{17}\text{O}$, $\delta^{17}\text{O}$, $\delta^{18}\text{O}$, we have for Park Forest glass, -0.33, 8.85, 17.40; and for irghizites -0.174, 6.63, 12.87 (avgs.).

The figure plots the composition of the Park Forest meteorite as an L5 chondrite with positive $\Delta^{17}\text{O}$ and with the Park Forest Glass as negative $\Delta^{17}\text{O}$. Over 30kg of the meteorite was recovered, analyzed and classified by the University of Chicago group as an L5 ordinary chondrite (see: Meteoritics & Planetary Science 39, Nr 4, 625-634 (2004)).

Please note that the observed fall in the Park Forest suburb was an L5 ordinary chondrite with positive $\Delta^{17}\text{O}$ but the Park Forest glass had a negative $\Delta^{17}\text{O}$. On this basis, I must question the authors' conclusion that the unrecovered Zhamanshin meteorite was a carbonaceous chondrite with negative $\Delta^{17}\text{O}$.

The reason I never published the results on Park Forest glass was that I couldn't propose a plausible explanation for the negative $\Delta^{17}\text{O}$ of the glass vs. the positive $\Delta^{17}\text{O}$ of the ordinary chondrite. I suspected an interaction between meteorite and atmospheric O₂ as it has been known since the work of Boaz Luz (Nature 1999) that atmospheric O₂ has a negative $\Delta^{17}\text{O}$ but I couldn't think of a physical mechanism to accomplish isotope exchange.

The plotted graph, drawn at the time the analyses were made, shortly after the fall of Park Forest, shows comparisons between the O-isotope compositions of Park Forest Glass, the L5 ordinary chondrite, and published data on fusion crusts on meteorites, and extra-terrestrial deep sea spherules. Many of the crusts and spherules show negative $\Delta^{17}\text{O}$.

I think the Park Forest data with a recovered ordinary chondrite with positive $\Delta^{17}\text{O}$ associated with "tektite"-like forms having negative $\Delta^{17}\text{O}$ strongly challenges the authors' conclusion that the unrecovered Zhamanshin meteorite was a carbonaceous chondrite. The plot of fusion crusts and deep-sea spherules, many of them with negative $\Delta^{17}\text{O}$, suggests a possible interaction between ballistic bodies heated by passage through the atmosphere and the atmosphere with its negative

$\Delta^{17}\text{O}$. The problem is this: the O-isotope compositions of secondary products of the passage of meteorites through the atmosphere, eg irghizites, Park Forest glass, fusion crusts, and deep sea spherules, do not appear to be related to the O-isotope composition of their meteorites.

I urge the authors to reconsider their interpretation of the unrecovered Zhamanshin meteorite as a carbonaceous chondrite. Their O-isotope data is not definitive. I would be willing to discuss with the authors a collaborative use of the unpublished data if it would be of service in achieving an improved interpretation.

Reviewer #1 (Remarks to the Author):

The Zamanshin impact glass has been demonstrated to contain a significant amount of extraterrestrial material (chondritic material in the 1-10% range) based on Ni, Co, Cr contents (Jonasova et al. 2016). These authors suggested a carbonaceous chondrite impactor possibly CB-CH type due to the high Ni/Cr ratio.

The present paper by the same group proposes to further constrain this parent body classification using two isotopic systems, oxygen and chromium. In both systems an ET signature seems demonstrated, obviously with Cr, likely with O as the alternative of an air contamination is difficult to account for. Nevertheless the shift in $\Delta^{17}\text{O}$ is minor: -0.1‰, i.e. in the range of crustal rocks.

Both isotopic results point toward a carbonaceous chondrite like impactor, confirming Jonasova et al. paper. This is worth publishing, possibly in a more specialized journal as these results do not lead to deeply innovative conclusions.

We are confident that the finding of a terrestrial material with such extreme Cr isotope systematics, though originating from contamination by an impactor, is highly significant in itself because previous findings of $\epsilon^{54}\text{Cr}$ values >1.5 were solely related to meteorites and not impact-related materials where the major mass of material originated from the Earth. A recent study by Mougél et al. (EPSL 2017) has provided further evidence that $\epsilon^{54}\text{Cr}$ values outside common mantle values may originate from impactors with distinct $\epsilon^{54}\text{Cr}$ values, with clues to their quantitative and qualitative estimates.

The remark about oxygen is important. It is correct that the shift in $\Delta^{17}\text{O}$ of ~ -0.1 to -0.2 ‰ is minor. However, combined with data for other tektites and impact-related glasses (published, see Fig. 1; unpublished, see review #3), it appears to have implications for a common process of interaction between impact-related materials and ambient atmosphere during the impact process. Moreover, the vector of the shift is not directed in crustal range (see our Fig. 1).

However, the way the isotopic couple is used to propose a Tagish lake like impactor denotes a major misunderstanding of what two component mixtures produce on isotopic systems. In particular the idea (implicit in the paper and Fig.3) that in a two isotopic diagram the mixture should plot on a straight line between the two end-members is plain wrong.

This is correct. Considering other oxygen isotope constraints (see above), we have re-structured the discussion with a more emphasis on the partial exchange with ambient air. This is also supported by electron microprobe data for irghizites. While in most cases of distinct oxygen isotope compositions of ET materials the mixing trajectory would be curved, for $\Delta^{17}\text{O}$ values in a narrow range of broadly Earth-like compositions, the trajectory would be more or less a straight line. Nevertheless, we have removed this from Fig. 3, as our explanation now favors other process to occur.

Let's precise the problem:

Trace elements have set an upper limit of ET contamination of 10% in mass. As a chondrite have an oxygen content only slightly less than a standard crustal rock, the amount of ET oxygen in the irghizite is less than 10%, say 5%. The 0.1 shift in thus translates into a $\Delta^{17}\text{O}$ of the impactor below -2 ‰. This definitely excludes a Tagish lake like parentage and points toward CB or other carbonaceous chondrite parentage.

Although the $\epsilon^{54}\text{Cr}$ systematics of meteorites are based on a rather small number of individual meteorites for each group, the available data show a clear distinction between the meteorite classes and also between individual types of carbonaceous chondrites. The high $\epsilon^{54}\text{Cr}$, found for one irghizite in this study, allows us to exclude the majority of carbonaceous chondrite types, also with implications for the interpretations given in Jonasova et al. (GCA 2016). Here, we also note that similar

conclusions on carbonaceous chondrites were reached for other impact sites by Mougél et al. (EPSL 2017). The $\Delta^{17}\text{O}$ values of ~ -0.1 – -0.2% , combined with high $\epsilon^{54}\text{Cr}$, delimit the range of possible chondrite types fairly well. From these combined data, no chondrite type with $\Delta^{17}\text{O} \sim -1\%$ can represent the viable impactor because it would violate Cr isotope constraints. Therefore, we point the discussion toward interaction with ambient atmosphere, which is a process yet unaccounted for in terrestrial rocks and which could be consistent with the unpublished data of reviewer #3 for other impact-related materials.

Concerning the Cr content, Jonasova results imply that a large majority of Cr in irghizite is of ET origin, thus the $\epsilon^{54}\text{Cr}$ will be close to the parent body value. The 1.5 value seems to point toward TL, but the possibility that it could fit with other CC groups is dismissed by the wrong visual impression that both oxygen and Cr values should fit the parent body!

At present, there are very few groups of carbonaceous chondrites which have such high $\epsilon^{54}\text{Cr}$ values. In fact, only CI chondrites and Tagish Lake chondrite appear to have such a high ^{54}Cr isotope value. We also do not implicitly state that the Cr–O systematics should be broadly consistent with the existing groups of carbonaceous chondrites but indirect lines of evidence appear to be consistent with present knowledge of this isotope tandem. Moreover, the current data set may provide evidence for a partial decoupling of Cr–O isotope systematics in an impact process.

What is also missing is a discussion of the variability of $\epsilon^{54}\text{Cr}$ in CC and the possible effect of impact volatilization on $\epsilon^{54}\text{Cr}$. It would have been clever to obtain data on moldavite to assess this effect as the present paper propose moldavite as a clear exemple of impact glass devoid of ET contamination. We have been able to collect Cr isotope data for one large enough specimen of a moldavite for which oxygen isotope composition was also measured in this study. The data unequivocally provides a direct evidence for no addition of Cr to moldavites, at least unresolvable with current analytical techniques. This supports our notion of lacking effects of volatilization because moldavites are known to have experienced higher processing temperatures.

A further example of major reasoning flaws can be found in the following sentence:

“A tight correlation of $\epsilon^{54}\text{Cr}$ versus $\Delta^{17}\text{O}$ for irghizite » where is this correlation? When one has only two data points how can one put forward such a claim ?!

This is correct and has already been discussed in above points. We made this statement on the basis of homogeneous ^{54}Cr in Earth which has been clearly shown in several studies utilizing Earth's materials, including a reference Hawaiian basalt from this study. This claim of Cr isotope homogeneity has now been confirmed by our new analysis of one moldavite specimen from the current sample suite for which oxygen isotope data was also acquired. For the sake of clarity, we have avoided the statement about the correlation and invoke that other processes could have partially modified oxygen isotope systematics.

Reviewer #2 (Remarks to the Author):

Issaku Kohl: review of NCOMMS-16-20902-T

Zhamanshin astrobleme: O–Cr isotope evidence for a carbonaceous chondrite impactor

Summary of Approach

The authors suggest that O and Cr (although much less Cr data is presented) isotope fingerprinting can identify impactor assimilation in tektite and similar type impact glasses.

Oxygen and chromium isotopic compositions of extraterrestrial materials have been extensively characterized and expected ranges for different impactor types are previously documented. Given well-characterized target rock lithologies and isotopic compositions, this is a valid approach.

This approach is applied to impact glasses thought to be associated with 2 impact structures, Zhamanshin (irghizites and more basic splash forms) and Ries (moldavites). The data shows significantly negative $\Delta 17\text{O}$ (twice as negative as terrestrial rocks) values for irghizites, while the basic splashforms from Zhamanshin and the moldavites show nominally terrestrial values.

General Comments and Suggested Changes

The authors effectively discount atmospheric oxygen assimilation as a possible source for the $\Delta 17\text{O}$ anomaly as it requires incorporation of an unreasonable amount of $\Delta 17\text{O}$ negative and $\delta 18\text{O}$ enriched air O_2 (-0.475‰ and 23.8‰ respectively). They also suggested that the terrestrial $\Delta 17\text{O}$ bearing basic splash forms and moldavites are simply lacking in impactor assimilation, appealing to the two rock types sampled by the Zhamanshin impact having different sampling depths. While not explicitly stated I believe the authors are appealing to depth segregation limiting interaction between impactor and basic target rocks for Zhamanshin. If that is the case, it should be stated explicitly. As it stands, there is no good explanation for acidic tektites bearing impactor related anomalies and basic ones not, this is a crucial observation that must be addressed

We have included the information about different chemistry of sources of irghizites and basic splash-forms with references where this information is detailed. However, with regard to the comments by reviewer #3 and considering his unpublished data, the oxygen isotope exchange with air may indeed be supported by our data. This will also lead to modest decoupling of O and Cr isotope systematics.

A brief discussion of siderophile element content and Cr isotope composition in irghizites allow the authors to discount major meteorite classes as possible impactors settling on a Tagish Lake type ungrouped chondrite as the most likely candidate. This needs to be highlighted in a more aggressive way. I believe this to be the most significant contribution in this manuscript. Currently, it reads as though the non-anomaly bearing tektites are the minority and much time and effort is put toward explaining why they have no anomaly. However, given that 2 out of 3 types of tektites (the majority) in this manuscript have no anomaly it seems pertinent to start with that as the “easy to explain case” and then go into the “special case” where an anomaly is inherited. This adds weight and sets the reader up for the ungrouped chondrite conclusion, which is the major contribution.

We have re-structured the discussion although the original intention was to highlight the major finding of irghizites being affected by the impactor mater compared to the other non-anomaly impact materials. Unfortunately, the revised version of our manuscript on siderophile element systematics in moldavites (moderate revisions were requested) is still under review for nearly two months and we cannot use this as support. We have also sized down our arguments on oxygen isotope signature to be derived from impactor.

In addition, I believe a short discussion on evaporation is needed. The authors allude to it in the last paragraph with one reference. For “impact folks” the formation of evaporative vapor condensates is a given and we know that the isotope effects will differ for vapor and liquid. I am not suggesting that this process plays a major role but I would like to see it receive some treatment. A few calculations showing a more quantitative attempt to model the potential role of evaporation, even in the supplement, would add to the thoroughness.

With this study to be released we hope to promote further experimental investigations. However, considering a large diversity in impactor types and sizes, target lithologies and local conditions, any generalization should be omitted. We have included a study by Engrand et al. (GCA2005) where the effects of partial exchange with air were advocated for silicate spherules. Such a kind of studies, however, should utilize high-resolution in-situ instrumentation combined with carefully controlled experiments, which was not available to us. We admit, though, that effects of this kind (i.e., evaporation) may appear if deconvolved with proper analytical tools.

Recommendation

Aside from the above suggestions, the manuscript is well written and the data look excellent. As to reproducibility of the data set, there are only a few labs in the world that are doing $\Delta 17O$ at the level of the Pack group at Gottingen, so for those other groups it should be attainable. All that said, I believe this is a good approach to identifying elusive impactor isotopic and chemical compositions from preserved impact glass sediments. It will benefit from some reorganization and a bit of additional information but I recommend this for publication with significant changes.

Best,

Issaku Kohl Ph.D.

Project Scientist II
Young Stable Isotope Laboratory
595 Charles E. Young Drive East
2676 Geology Building
Los Angeles, CA, 9095
ikohl@epss.ucla.edu

Reviewer #3 (Remarks to the Author):

Please see text, table, and figure in the uploaded DOC file.

Thanks to the editor for the opportunity to review this most interesting submission to Nature Communications. The work is of particular interest because it reports data strongly reminiscent of samples taken from the rain gutters of houses in the Chicago suburb of Park Forest shortly after the fall of a large meteorite on 26 March 2003. The samples were collected by meteorite collectors and sent to me for O-isotope analysis by Tony Irving (Univ. Washington). I'm not writing a conventional ms. review because I think data from Park Forest may help the authors of the ms. to reach a better interpretation of the origin of the aerodynamically-shaped glassy forms found at Zhamanshin.

We are grateful for these 'unconventional review' remarks. We note here that our interpretation of the impactor being a carbonaceous chondrite is not based solely on O isotope data but was much aided by employing Cr isotopes. This isotope system appears to unequivocally distinguish between various meteorite types, irrespective of O isotope compositions. Notably, the Earth's Cr systematics are well-constrained and cannot be modified without a significant input of extra-terrestrial Cr. In order to test this, we have analyzed one tektite from this study for Cr isotope systematics. A clearly terrestrial value was obtained. Moreover, modeling has shown consistency for both ^{54}Cr and ^{53}Cr to derive from an impactor and variable lithologies for impact-related glasses from Zhamanshin. This has now been depicted in new Fig. 4.

We are confident that O isotope systematics could result from partial interaction with ambient atmosphere which has now become our working hypothesis. This process could also be noted for other groups of tektites because they show a trend toward more negative $\Delta^{17}\text{O}$ values compared with common terrestrial lithologies. The findings of Reviewer #3 for his samples underscore this interpretation. We recollect that this partial interaction will not influence the Cr systematics and our interpretation of the impactor as a carbonaceous chondrite will remain valid.

When I received the samples from Tony Irving, I was astonished to see, under a binocular microscope, glassy mm-sized aerodynamic forms. Dumbbell shapes were common with two bulbous ends separated by a narrow connecting filament. There were also half-dumbbells with a single bulbous end and a filament tapering to a sharp end. As I recall, there was a single example of a specimen that had its filament folded back upon itself giving the shape of a jug with a curved handle. I can't send photos because I'm in Paris, giving a seminar at I'IPGP on Friday, and can't access records stored in my lab in Washington, DC.

Above, I have copied analytical data on the O-isotopes of the dumbbells. In the data table, duplicate analyses of the same gas sample are given for run 12-4. The other runs, 11-204 and 12-9 are single analyses of samples fluorinated at different times. I used BrF_5 as fluorinating reagent and heated the samples to promote fluorination with a CO_2 laser and calculated $\Delta^{17}\text{O}$ with the factor 0.526. As can be seen in the data, the Park Forest dumbbells are remarkably similar to the O-isotope data of the irghizites: In the order $\Delta^{17}\text{O}$, $\delta^{17}\text{O}$, $\delta^{18}\text{O}$, we have for Park Forest glass, -0.33, 8.85, 17.40; and for irghizites -0.174, 6.63, 12.87 (avgs.).

The figure plots the composition of the Park Forest meteorite as an L5 chondrite with positive $\Delta^{17}\text{O}$ and with the Park Forest Glass as negative $\Delta^{17}\text{O}$. Over 30kg of the meteorite was recovered, analyzed and classified by the University of Chicago group as an L5 ordinary chondrite (see: Meteoritics & Planetary Science 39, Nr 4, 625–634 (2004)).

Please note that the observed fall in the Park Forest suburb was an L5 ordinary chondrite with positive $\Delta^{17}\text{O}$ but the Park Forest glass had a negative $\Delta^{17}\text{O}$. On this basis, I must question the authors' conclusion that the unrecovered Zhamanshin meteorite was a carbonaceous chondrite with negative $\Delta^{17}\text{O}$.

The reason I never published the results on Park Forest glass was that I couldn't propose a plausible explanation for the negative $\Delta^{17}\text{O}$ of the glass vs. the positive $\Delta^{17}\text{O}$ of the ordinary chondrite. I suspected an interaction between meteorite and atmospheric O_2 as it has been known since the work of Boaz Luz (Nature 1999) that atmospheric O_2 has a negative $\Delta^{17}\text{O}$ but I couldn't think of a physical mechanism to accomplish isotope exchange.

The plotted graph, drawn at the time the analyses were made, shortly after the fall of Park Forest, shows comparisons between the O-isotope compositions of Park Forest Glass, the L5 ordinary chondrite, and published data on fusion crusts on meteorites, and extra-terrestrial deep sea spherules. Many of the crusts and spherules show negative $\Delta^{17}\text{O}$.

I think the Park Forest data with a recovered ordinary chondrite with positive $\Delta^{17}\text{O}$ associated with "tektite"-like forms having negative $\Delta^{17}\text{O}$ strongly challenges the authors' conclusion that the unrecovered Zhamanshin meteorite was a carbonaceous chondrite. The plot of fusion crusts and deep-sea spherules, many of them with negative $\Delta^{17}\text{O}$, suggests a possible interaction between ballistic bodies heated by passage through the atmosphere and the atmosphere with its negative $\Delta^{17}\text{O}$. The problem is this: the O-isotope compositions of secondary products of the passage of meteorites through the atmosphere, eg irghizites, Park Forest glass, fusion crusts, and deep sea spherules, do not appear to be related to the O-isotope composition of their meteorites.

I urge the authors to reconsider their interpretation of the unrecovered Zhamanshin meteorite as a carbonaceous chondrite. Their O-isotope data is not definitive. I would be willing to discuss with the authors a collaborative use of the unpublished data if it would be of service in achieving an improved interpretation.

Reviewers' Comments:

Reviewer #1:

Remarks to the Author:

I am happy to see that the authors have strongly amended their interpretation of the oxygen data and understood that their previous two isotopic system interpretation with same end-members (terrestrial target and meteorite) was not correct.

Interpretation of the data (with welcome addition, e.g. ^{54}Cr of moldavite) is now sound and I would recommend to accept the paper with minor revision according to the comments below.

a) ^{54}Cr signal of 1.54 ± 0.08 for IR8 is said to exclude all known meteorites except CI and TL; however, based on the welcome added reference of Mougel et al., it seems that the average signal for CR is not significantly different: 1.48 ± 0.25 (one sigma).

Interestingly, in the hypothesis that oxygen shift is due to mixing with ET material, CR would do the job !

b) Still I think the mixing with atmospheric oxygen is the right answer for the large delta shift ; this mixing is clearly evidenced by the richness in Fe^{3+} of irghizite (as discussed in the paper). The absence of shift for the moldavite is accounted for by a brief and unclear sentence : « The model of slow gravitational settling and chemical and isotope exchange with a slowly collapsing vapor plume cannot be applied to moldavites (distal ejecta) due to their instantaneous dislocation from the area of origin. ». A more simple point that could be made is that minimal exchange with air is indicated by the lack of Fe^{3+} in moldavite (see review on iron oxidation state of tektite and impact glass in Rochette et al. 2015, EPSL 432, 381).

c) « this makes irghizites of the Zhamanshin astrobleme one of the few materials in the geological record that carries traces of the isotope anomaly of air O_2 . « I would mention the case of cosmic spherules, that shows similar shift due to the same process (e.g. Suavet et al. 2010, EPSL 293, 313)

Reviewer #2:

Remarks to the Author:

I am very happy with the changes made to the manuscript. Both in response to my comments and those of the other reviewers.

The separation achieved by the using Cr isotopes to identify impactor type, followed by significant vs insignificant atmospheric contamination simplifies the story.

In the improved discussion the authors addressed all of my concerns effectively and demonstrate why and how the decoupling between Cr and O could occur. I still wish there was more Cr isotope measurements but that can't be helped, i guess.

I have one small (but important) suggestion concerning lines 52-57 where solar system mass independent fractionation is introduced and atmospheric oxygen's ^{17}O depletion is discussed.

In these sentences comparing solar system bodies to mass fractionation within a single body (Earth) is not really appropriate. Within any given body, processes are mostly mass dependent (e.g. Mars). The large differences between bodies are inherited from mass independent processes attending early solar system formation.

Specifically addressing the cause of the anomaly in air O_2 . In Young et al, (2014, ref .7) it is made clear that the majority of the ^{17}O depletion in air is in fact the result of biological mass dependent processes with smaller Beta values (relative to rocks and waters) and mostly not related to photo chemistry (which does contribute some). I know this is secondary to the goal of the paper but it should still be addressed for accuracy.

Once the above small change is made I believe this paper is ready for publication.
Thank you,
Issaku Kohl

Reviewer #1 (Remarks to the Author):

Interpretation of the data (with welcome addition, e.g. ^{54}Cr of moldavite) is now sound and I would recommend to accept the paper with minor revision according to the comments below.

a) ^{54}Cr signal of 1.54 ± 0.08 for IR8 is said to exclude all known meteorites except CI and TL; however, based on the welcome added reference of Mougél et al., it seems that the average signal for CR is not significantly different: 1.48 ± 0.25 (one sigma). Interestingly, in the hypothesis that oxygen shift is due to mixing with ET material, CR would do the job!

From the current data set, we cannot exclude a CR chondrite as an impactor. This is also discussed in the main text where a simple calculation is not at odds with CR-type chondrite impactor. Considering the large uncertainty in ^{54}Cr isotope data for most carbonaceous chondrite groups, this issue clearly demands further careful work on different classes of meteorites, which may provide the ultimate precision for individual groups of chondrites and other extraterrestrial materials.

b) Still I think the mixing with atmospheric oxygen is the right answer for the large delta shift; this mixing is clearly evidenced by the richness in Fe^{3+} of irghizite (as discussed in the paper). The absence of shift for the moldavite is accounted for by a brief and unclear sentence: « The model of slow gravitational settling and chemical and isotope exchange with a slowly collapsing vapor plume cannot be applied to moldavites (distal ejecta) due to their instantaneous dislocation from the area of origin. ». A more simple point that could be made is that minimal exchange with air is indicated by the lack of Fe^{3+} in moldavite (see review on iron oxidation state of tektite and impact glass in Rochette et al. 2015, EPSL 432, 381).

This is a circular argument. We have stated that tektites represent a high-velocity distal ejecta that were dislocated from their place of origin almost instantaneously. This rapid process has prevented them to isotopically exchange with ambient air whilst fallback particles, such as irghizites, have stayed in contact with local atmosphere for an extended period of time. The lack of ferric iron in moldavites is a result of redox reactions during volatile loss from tektite melts because much of original iron in parental sediments was present in form of ferric iron.

c) « this makes irghizites of the Zhamanshin astrobleme one of the few materials in the geological record that carries traces of the isotope anomaly of air O_2 . « I would mention the case of cosmic spherules, that shows similar shift due to the same process (e.g. Suavet et al. 2010, EPSL 293, 313) We have inserted “terrestrial” to make it clear that we do not consider extraterrestrial materials in this place and that we focus on materials of terrestrial origin here which carry traces of air O_2 .

Reviewer #2 (Remarks to the Author):

I am very happy with the changes made to the manuscript. Both in response to my comments and those of the other reviewers. The separation achieved by the using Cr isotopes to identify impactor type, followed by significant vs insignificant atmospheric contamination simplifies the story. In the improved discussion the authors addressed all of my concerns effectively and demonstrate why and how the decoupling between Cr and O could occur. I still wish there was more Cr isotope measurements but that can't be helped, I guess.

The Cr isotope analysis of this type is much more demanding than a more conventional stable Cr isotope analysis. We hope that future studies of most relevant samples may provide further constraints on the impactors in other cases. Even now, low Cr contents made the analysis of the selected “Cr-rich” moldavite challenging.

I have one small (but important) suggestion concerning lines 52-57 where solar system mass independent fractionation is introduced and atmospheric oxygen's ^{17}O depletion is discussed. In these sentences comparing solar system bodies to mass fractionation within a single body (Earth) is not really appropriate. Within any given body, processes are mostly mass dependent (e.g. Mars). The large

differences between bodies are inherited from mass independent processes attending early solar system formation.

This is a correct remark. We have modified the statement accordingly.

Specifically addressing the cause of the anomaly in air O₂. In Young et al, (2014, ref .7) it is made clear that the majority of the ¹⁷O depletion in air is in fact the result of biological mass dependent processes with smaller Beta values (relative to rocks and waters) and mostly not related to photo chemistry (which does contribute some). I know this is secondary to the goal of the paper but it should still be addressed for accuracy.

This is a correct and important remark. We have modified the statement accordingly.